# Self-Dual: Unifying Natural Language and Programmatic Thinking for Enhanced Mathematical Reasoning in LLMs

## Abstract

Large language models (LLMs) have made significant progress in mathematical reasoning. However, the methods that rely on a single reasoning paradigm exhibit clear limitations. This has motivated recent studies to combine multiple paradigms, but existing studies often fail to systematically exploit their complementary strengths. In this study, we first examine the complementary relationship between natural language (NL) and programmatic language (PL) reasoning, and show that their integration leads to consistent improvements in mathematical reasoning performance. Building on this analysis, we introduce Self-Dual, a framework that unifies the two paradigms within a single inference process by generating complementary reasoning trajectories and combining them through structured self-reflection. Beyond inference, we extend this principle to training by adopting the Self-Dual data format to construct complementary reasoning datasets and evaluate its effectiveness in model training. We conduct comprehensive evaluations of Self-Dual in both inference and training contexts. During inference, Self-Dual consistently surpasses NL-only, PL-only, and hybrid baseline methods across multiple benchmarks. DeepSeek-V3-0324 integrated with Self-Dual attains 47.8% accuracy on the AIME25 dataset, outperforming Chain-of-Thought (CoT) at 39.2% and Program-Aided Language (PAL) at 35.6%. In the training experiments, we apply the Self-Dual framework to further train Qwen2.5-7B-Instruct with only 7.5K MATH samples and construct Qwen2.5-7B-SD. The new model improves performance on MATH500 by more than 4% over the base model Qwen2.5-7B-Instruct. It also surpasses Qwen2.5-Math-7B-Instruct on AIME25. These results demonstrate that the Self-Dual framework effectively exploits complementary reasoning paradigms and substantially enhances the mathematical reasoning ability of large language models in both inference and training.

## 1 Introduction

Large Language Models (LLMs) (Achiam et al., 2024; Guo et al., 2025; Qwen et al., 2025) make remarkable strides across a diverse array of tasks through the application of prompting techniques. Nevertheless, complex reasoning tasks (Xie et al., 2024; Sprague et al., 2024) such as mathematical reasoning continue to present significant challenges. NL-based prompting methods, such as Chain-of-Thought (CoT), utilize natural language to decompose complex reasoning tasks into multiple intermediate steps and solve them step by step (Wei et al., 2022; Zhou et al., 2023; Li et al., 2023). These methods offer high interpret-ability but are prone to calculation and logical errors (Madaan & Yazdanbakhsh, 2022; Nogueira et al., 2021; Qian et al., 2022). In contrast, PL-based prompting methods like Program-aided Language (PAL) (Gao et al., 2023b) generate programs as intermediate reasoning steps and offload the solution steps to a runtime such as a Python interpreter. They are more logically robust and computationally accurate (Chen et al., 2023; Gou et al., 2024b). However, such methods often struggle to represent certain steps for solving natural language problems in programming language.

Prior work on combining NL-based and PL-based reasoning can be grouped into two categories. At inference time, methods such as Automatic (Zhao et al., 2023) select the final answer from CoT and PAL outputs, but are sensitive to candidate ordering and often unstable. Other methods,

like CRITIC (Gou et al., 2024a), refine programs through NL reflection, yet remain limited by the expressiveness of PL. At training time, approaches such as MathCoder (Wang et al., 2023) construct datasets by deriving formulas and code with PL reasoning, while SAAS (Kim et al., 2024) and Qwen2.5-Math (Yang et al., 2024a) mix NL and PL data for joint training. CoR (Yu et al., 2025b) adopts progressive paradigm training, where the model is first trained on NL-reasoning data and then on PL-reasoning data. All these methods empirically combine NL-based reasoning and PL-based reasoning, but they overlook the central principle of complementarity between them, leaving much of their combined potential unexplored.

In this paper, our observation confirms that NL reasoning and PL reasoning are inherently complementary: their union consistently outperforms either alone, while their intersection reflects convergence across reasoning attempts, as shown in Figure 1. This suggests that hybrid reasoning should explicitly leverage complementarity, rather than treat NL and PL as independent or sequential signals. Building on this insight, we propose Self-Dual, a unified framework that exploits the complementarity of NL and PL reasoning to enhance inference, while also serving as a data construction format to support model training. At inference time, Self-Dual produces two complementary reasoning paths within a single forward pass: one in NL and one in programmatic form. These paths are then compared and integrated through a structured self-reflection process consisting of three steps: Look Back, Decomposition, and Resolution. At training time, Self-Dual generates complementary trajectories that are used as cold-start data. These data are then integrated into Group Relative Policy Optimization (GRPO) (Shao et al., 2024) with a tailored reward function to assess whether the Self-Dual format can improve model performance in mathematical reasoning.

For inference-time experiments, we evaluate the empirical performance of Self-Dual across multiple series of LLMs, including DeepSeek-V3-0324 (DeepSeek-AI, 2024) and Gemma 3 (Team, 2025) variants (4B, 12B, 27B), on diverse mathematical reasoning benchmarks. Our findings show that Self-Dual consistently surpasses prior techniques without supplementary data or training. For example, DeepSeek-V3-0324 with Self-Dual BAND achieves 4% improvement in accuracy on the MATH500 benchmark and 16.66% gain on AIME25. For training-time experiments, we use Qwen2.5-7B-Instruct (Qwen et al., 2025) as the base model and train it with Self-Dual data, obtaining Qwen2.5-7B-SD. On AIME25, Qwen2.5-7B-SD surpasses Qwen2.5-Math-7B-Instruct (Yang et al., 2024a), which is trained with large-scale hybrid data and GRPO. This result shows that the Self-Dual format can deliver competitive improvements in mathematical reasoning with much smaller training data.

## 2 RELATED WORK

### 2.1 ENHANCING MATHEMATICAL REASONING

There are two main techniques for enhancing the mathematical reasoning capabilities of LLMs: prompting methods and fine-tuning-based methods. The in-context few-shot learning through prompting is simple and broadly applicable, like popular CoT (Wei et al., 2022) and PAL (Gao et al., 2023b). Automatic-CoT (Zhang et al., 2023), Many-Shot (Agarwal et al., 2024) and Synthetic Prompting (Sprague et al., 2024) explore improving model performance by enhancing the quality or quantity of few-shot examples. ToT (Yao et al., 2023), GoT (Besta et al., 2024) and BoT (Yang et al., 2024b) design different ways of structuring thoughts to further enhance LLMs performance. TIR (Gou et al., 2024b), CRITIC (Gou et al., 2024a) and Recursive-Introspection (Qu et al., 2024) perform multi-round self-evaluation based on previous results to iteratively improve the reasoning process. The most related to our work are Mathprompter (Kim et al., 2023a) and Automatic (Zhao et al., 2023), which use simple selection or majority voting to select the results of PAL and CoT. Unlike these methods, our approach decomposes the complementary reasoning paths to extract their respective strengths and synthesizes a new solution during the refine stage.

Beyond prompt-based approaches, fine-tuning improves mathematical reasoning by training on curated datasets (Tong et al., 2024a; Zhou et al., 2024; Luo et al., 2025a) or large-scale synthetic data, such as MetaMath (Yu et al., 2024) and DeepSeekMath (Shao et al., 2024). Building on this line, recent work explores multi-paradigm reasoning by mixing natural language, code, and symbolic forms (Yu et al., 2025a; Zheng et al., 2025) or by integrating reinforcement learning and external tools (Luo et al., 2025a; Gou et al., 2024b). Yet these methods treat NL-based and PL-based rea-

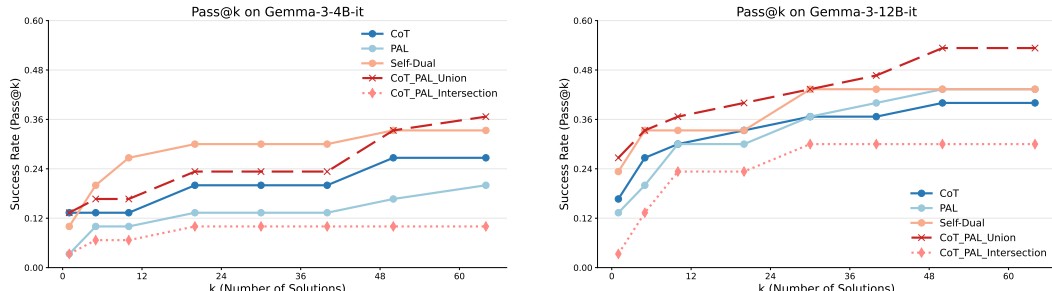

Figure 1: Pass@k curves of Gemma3-4B-it and Gemma3-12B-it on CoT, PAL, Self-Dual, as well as the union and intersection of CoT and PAL, evaluated on the AIME25 dataset. Setup is given in Section 4.1.1.

soning as simple combinations, overlooking their complementarity. Our work instead unifies both paradigms within a single inference pass and reconciles them through self-reflection.

## 2.2 SELF-REFINE

Self-Refine refers to the ability of LLMs to improve or revise their previous responses based on feedback (Madaan et al., 2023; Shinn et al., 2023). Automated feedback has primarily been used for training-time correction (Yan et al., 2023) or for guiding LLMs during inference through prompting (Weng et al., 2023). One crucial aspect of self-refinement is that the source of feedback provides reliable and effective additional information. A vast amount of research leverage external feedback for optimization, including external knowledge sources (Gao et al., 2023a), external tools (Chern et al., 2024), program executors (Chen et al., 2024), symbolic solvers (Pan et al., 2023), or even trained models (Paul et al., 2024). Some approaches, such as Self-Refine (Madaan et al., 2023) and RCI (Kim et al., 2023b), further optimize responses through multiple iterations by leveraging the model's inherent reflection capabilities. While the primary goal of our method is to generate complementary solutions that inspire reflection, external feedback from the Python interpreter is also integrated into the refine stage as a component of the PL-based approach.

## 3 METHODOLOGY

### 3.1 COMPLEMENTARITY BETWEEN REASONING MODES

NL-based methods are adept at capturing semantic and intuitive reasoning, while PL-based methods primarily emphasize symbolic and structured reasoning. Intuitively, integrating these two paradigms is expected to enhance reasoning capabilities, much like the human cognitive process that combines intuition with logic. Although several recent studies (Yu et al., 2025b) have explored the integration of NL and PL to improve the reasoning ability of LLMs, to the best of our knowledge, a systematic understanding of their complementary interplay is still missing.

In this work, we explicitly examine the notion of complementarity and conceptualize it as a central principle underlying the effectiveness of hybrid NL–PL reasoning. Concretely, we position CoT as the representative paradigm of NL-based reasoning and PAL as the representative paradigm of PL-based reasoning. By analyzing these two paradigms, we provide empirical evidence supporting their inherent complementarity.

To evaluate performance, we adopt the Pass@K metric, which measures the probability of obtaining at least one correct solution when sampling $K$ diverse outputs from a model. Formally, let $S_i$ denote the set of solutions generated by method $i$ ($i \in \text{CoT}, \text{PAL}$), and let $\mathcal{C}$ denote the set of correct solutions. Then Pass@K is defined as:

$$\text{Pass@K}(S_i) = \mathbb{P}\big[S_i \cap \mathcal{C} \neq \emptyset\big]. \tag{1}$$

In addition to evaluating each method individually, we also analyze the union and intersection of the solutions produced by CoT and PAL. The union reflects the expanded solution space when

both paradigms are combined, while the intersection reveals their agreement. Examining these sets allows us to better understand the complementary and consistent aspects of NL-based and PL-based reasoning. They can be expressed as:

$$S_{\cup} = S_{\text{CoT}} \cup S_{\text{PAL}}, \quad S_{\cap} = S_{\text{CoT}} \cap S_{\text{PAL}}. \tag{2}$$

As illustrated in Fig. 1, our key observations are as follows.

**Union as upper bound.** Pass@K$(S_{\cup})$ increases steadily with $K$ and consistently surpasses both CoT and PAL alone. This demonstrates that NL-based and PL-based reasoning are complementary. Since their outputs are not fully overlapping, the union expands the effective solution set. Pass@K$(S_{\cup})$ therefore serves as an upper bound for hybrid reasoning.

**Intersection as consistency.** Pass@K$(S_{\cap})$ increases and gradually converges as $K$ grows. This behavior indicates that CoT and PAL exhibit increasing consistency in their outputs. At the same time, Pass@K$(S_{\cap}) \leq \min\big(\text{Pass@K}(S_{\text{CoT}}), \text{Pass@K}(S_{\text{PAL}})\big)$, which confirms that their solution sets are not fully overlapping. This motivates the use of the symmetric difference to characterize their non-shared components:

$$S_{\Delta} = (S_{\text{CoT}} \setminus S_{\text{PAL}}) \cup (S_{\text{PAL}} \setminus S_{\text{CoT}}) \tag{3}$$

**Symmetric difference as complementarity.** The non-empty symmetric difference further supports the claim that CoT and PAL capture distinct reasoning capacities. Their complementarity provides opportunities for hybrid methods to exploit.

In a nutshell, the union highlights the upper bound of hybrid reasoning, the intersection reflects growing consistency, and the symmetric difference confirms complementarity. Together, these observations provide a foundation for designing methods that explicitly exploit NL–PL synergy.

### 3.2 INFERENCE-TIME COMPLEMENTARITY

To exploit the complementary strengths of NL and PL reasoning, we propose Self-Dual, a hybrid framework that integrates both paradigms within a single inference. The key idea is to generate dual reasoning trajectories in one forward pass and then combine them through structured self-reflection. This design preserves the diversity of multiple reasoning paths while maintaining the efficiency of single-shot inference.

Self-Dual operates in two stages: Dual-Path Generation and Refinement. In the first stage, the model produces an NL-based reasoning path $R_{\text{NL}} = \text{LLM}(Q; \theta_{\text{NL}})$, followed by a PL-based path $R_{\text{PL}} = \text{LLM}(Q, R_{\text{NL}}; \theta_{\text{PL}})$ with $R_{\text{NL}}$ as context. This sequential generation improves consistency between the two paradigms, as discussed in Section 4.1.3. During the generation of $R_{\text{PL}}$, external feedback can be incorporated via stop tokens ""`output`", allowing partial code execution and verification $f(R_{\text{PL}}) \rightarrow$ Execution Feedback to strengthen the reliability of the PL trajectory.

In the second stage, Refinement, the two paths are compared and integrated in three structured steps. Look Back prompts the model to revisit the original problem statement $Q$, extract key information, and mitigate attention drift. Decomposition contrasts $R_{\text{NL}}$ and $R_{\text{PL}}$ to identify agreements and conflicts between the two paths. Resolution then consolidates $R_{\text{NL}}$, $R_{\text{PL}}$, and the execution feedback $f(R_{\text{PL}})$ into a refined trajectory $R$, from which the final answer $A = \text{Answer}(R)$ is derived and presented as "`boxed{A}`".

### 3.3 TRAINING-TIME COMPLEMENTARITY

At training time, we use the Self-Dual framework as a data generator to construct complementary reasoning trajectories. These trajectories serve as cold-start data for reinforcement learning and allow us to examine whether the Self-Dual format can enhance mathematical reasoning through data construction. We adopt standard GRPO for training and design a reward function that preserves the complementary reasoning pattern of Self-Dual:

$$R = \delta \cdot (\text{format reward}), \tag{4}$$

where $\delta \in \{0, 1\}$ is a binary coefficient, and format_reward is provided by an external reward model.

The value of $\delta$ is determined by two conditions. The first condition enforces a complementarity constraint: the model must output both an NL-based and a PL-based answer, preserving the reasoning format of Self-Dual during training. We do not evaluate the correctness of these intermediate answers, so the model can explore diverse reasoning paths. The second condition is the correctness of the final answer $A$, which determines the success of the solution. The coefficient $\delta$ is set to 1 only when both conditions are satisfied. The format reward term provides a fine-grained evaluation of reasoning quality, covering the completeness of the three-stage Self-Dual process, the correctness of the reasoning trajectory, and the correctness of the final answer.

## 4 EXPERIMENTS

### 4.1 INFERENCE-TIME

#### 4.1.1 SETUP

**Benchmarks.** We evaluate our approach on four widely-used benchmarks in the field of mathematical reasoning: SVAMP (Patel et al., 2021), GSM8K (Cobbe et al., 2021), MATH500 (Lightman et al., 2024) and AIME25. Their difficulty increases progressively in the given order. Please see Table 7 for details in Appendix.

**LLMs.** We conduct experiments using DeepSeek-V3-0324 (DeepSeek-AI, 2024), as well as the Gemma-3 Instruct model (Team, 2025) across three parameter scales: 4B, 12B, and 27B. To ensure a fair comparison, we set the temperature of all LLMs to 0.7 and the maximum token limit to 8192 for all experiments.

**Baselines.** We group the baselines into three categories. For NL-based methods, we consider CoT (Wei et al., 2022), which solves problems step by step in natural language, and Reflexion (Shinn et al., 2023), which follows the Response–Evaluation–Revision paradigm. For PL-based methods, we include PAL (Gao et al., 2023b), which generates executable programs. For hybrid-based methods, we evaluate Automatic (Zhang et al., 2023), a simple selection between CoT and PAL, Self-Dual-Automatic, which integrates independently executed CoT and PAL under the Self-Dual framework, CRITIC (Gou et al., 2024a), which iteratively refines programs using natural language feedback, and TIR (Gou et al., 2024b), which generates a rationale before producing and refining code. We set the maximum number of iterations to $n = 4$ for iterative methods, while for single-pass methods such as CoT and Self-Dual we report the average over three runs.

#### 4.1.2 MAIN RESULTS

In Table 1, we report both the accuracy and the number of API/LLM calls (#Call), which serves as an approximate measure of computational cost. Table 6 shows remaining results of GSM8K and SVAMP in Appendix. Hybrid-based methods consistently outperform single-paradigm methods across both benchmarks. While NL-based methods such as CoT and Reflextion capture semantic reasoning effectively, and PL-based methods such as PAL and TIR emphasize symbolic reasoning, their performance remains limited when used in isolation. By contrast, hybrid-based approaches achieve clear improvements, highlighting the intrinsic complementarity between NL-based and PL-based paradigms.

The proposed Self-Dual framework establishes better performance across all model scales, surpassing both automatic switching methods and their tool-free variants. Notably, Self-Dual achieves 47.78% on AIME25 with DeepSeek-V3-0324 and 90.04% on MATH500, demonstrating that explicitly generating and reconciling dual reasoning paths yields stronger gains than heuristic integration. These improvements are particularly pronounced for smaller models, where Self-Dual narrows the gap with much larger baselines. Tool feedback plays a crucial role in amplifying the benefits of programmatic reasoning. As shown by the "w/o tool" settings, removing Python interpreter feedback substantially degrades performance for PAL, CRITIC, and Self-Dual, confirming that external verifiability strengthens the PL-based reasoning pathway. Importantly, even without tool feedback, Self-Dual still maintains competitive accuracy, underscoring the robustness of its complementary reasoning design.

Table 1: Solve rates on AIME25 and MATH500. The "w/o tool" setting indicates that the method described in the previous line does not utilize the execution results from the Python interpreter as feedback. D-V3 refers to DeepSeek-V3-0324, while G-4B, G-12B, and G-27B denote different sizes of the Gemma-3 family. SD-Auto denotes the Self-Dual-Automatic method. The best results are highlighted in bold, and the second-best are underlined.

| Methods | AIME25 | | | | MATH500 | | | | |
|---|---|---|---|---|---|---|---|---|---|
| | **D-V3** | **G-4B** | **G-12B** | **G-27B** | **D-V3** | **G-4B** | **G-12B** | **G-27B** | **#Call** |
| *NL-based methods* | | | | | | | | | |
| CoT | 27.78 | 8.9 | 17.78 | 20 | 87.67 | 62.2 | 79.87 | 85.53 | 1 |
| Reflextion | 36.67 | 10 | 20 | 26.67 | 88 | 64.8 | 79.6 | 85.2 | 3 |
| *PL-based methods* | | | | | | | | | |
| PAL | 16.67 | 0 | 13.33 | 16.67 | 68 | 35.2 | 58.4 | 68.6 | 1 |
| *Hybrid-based methods* | | | | | | | | | |
| TIR | 13.33 | 0 | 6.67 | 23.33 | 81 | 43.4 | 63 | 77.2 | 1-5 |
| *w/o tool* | 13.33 | 0 | 3.33 | 10 | 76 | 18.8 | 32.8 | 45.8 | 1 |
| CRITIC | 13.33 | 3.33 | 10 | 10 | 68.4 | 39.8 | 60.6 | 68.6 | 9 |
| *w/o tool* | 16.67 | 3.33 | 10 | 16.67 | 65.6 | 37.6 | 60.6 | 65.2 | 9 |
| Automatic | 40 | 6.67 | 23.33 | 26.67 | 81.8 | 60.2 | 75.8 | 82.6 | 3 |
| SD-Auto | 40 | 10 | 23.33 | 26.67 | 88 | 67 | 82 | 85.6 | 3 |
| Self-Dual | **47.78** | **15.56** | **24.44** | **30** | **90.04** | **70.33** | **81.6** | **86.33** | 2 |
| *w/o tool* | 26.67 | 10 | 16.67 | 16.67 | 87.4 | 68 | 79.6 | 86 | 1 |

### 4.1.3 ANALYSIS

To further investigate the proposed framework in depth, we randomly sample 100 examples from MATH500. All sampled problems are kept consistent across methods to ensure a fair comparison. A comparative analysis of all methods is shown in Table 2, based on the consistency of the two initial reasoning paths and the correctness of the final outputs in the refine stage. We define four main categories: *Concordant Correctness (CC)*, *Concordant Error (CE)*, *Discordant Correctness (DC)*, and *Discordant Error (DE)*. *CC* denotes cases where both NL- and PL-based paths give the same correct answer, and refinement preserves it. *CE* corresponds to both paths producing the same wrong answer, which refinement follows. When the two paths are different, the case is Discordant. *DC* indicates that refinement yields the correct answer, while *DE* means it remains incorrect. We further distinguish $DC_{NL}$ and $DE_{NL}$ when the NL path is correct, and $DC_{PL}$ and $DE_{PL}$ when the PL path is correct. *Badcode* refers to PL programs that fail to execute, *FN (False Negative)* to correct answers mismatched with the ground truth, and *No2Paths* to cases where dual solutions are not generated.

Table 2: Results of Self-Dual, Self-Dual-Auto, Auto, and DCT on MATH500, categorized by the consistency of the initial reasoning paths and the correctness of the final outputs in the refine stage.

| Models | Methods | *CC* | *CE* | *DC* | *DE* | *Badcode* | *FN* | *No2Paths* |
|---|---|---|---|---|---|---|---|---|
| | Self-Dual | 77% | 4% | 6% | 4% | 4% | 1% | 4% |
| DeepSeek-V3-0324 | Self-Dual-Auto | 67% | 1% | 8% | 9% | 15% | 0% | - |
| | Auto | 67% | 1% | 4% | 13% | 15% | 0% | - |
| | Self-Dual | 67% | 6% | 8% | 5% | 4% | 1% | 9% |
| Gemma-3-12B-it | Self-Dual-Auto | 53% | 4% | 16% | 13% | 12% | 2% | - |
| | Auto | 53% | 4% | 11% | 18% | 12% | 2% | - |
| | DCT | 80% | 9% | 4% | 1% | - | 2% | 4% |

**Complementary Analysis.** In this paper, we regard the divergence between the NL-based and PL-based methods in the initial stage as the main concrete manifestation of their underlying complementarity. As shown in Table 2, in Gemma-3-12B-it, the proportion of Discordant outputs in

Table 3: Confidence of NL-based and PL-based methods when a divergence occurs in the initial stage.

| Models | Methods | $C_{NL}$ | $C_{PL}$ |
|---|---|---|---|
| DeepSeek-V3-0324 | Self-Dual | 50% | 30% |
| | Self-Dual-Auto | 41.18% | 27.65% |
| Gemma-3-12B-it | Self-Dual | 61.54% | 15.38% |
| | Self-Dual-Auto | 51.72% | 20.69% |

Table 4: Selection accuracy of each method in the refine stage.

| Models | Methods | *SCR* | *DCR* |
|---|---|---|---|
| DeepSeek-V3-0324 | Self-Dual | 95.4% | 60% |
| | Self-Dual-Auto | 89.29% | 47.06% |
| | Auto | 84.52% | 23.53% |
| Gemma-3-12B-it | Self-Dual | 93.75% | 61.54% |
| | Self-Dual-Auto | 84.15% | 55.17% |
| | Auto | 78.05% | 37.93% |

Self-Dual is higher than that of DCT. This indicates that employing NL and PL reasoning separately does indeed increase the likelihood of generating divergent outputs.

The comparison between Self-Dual and Self-Dual-Auto can be viewed as an evaluation of generating complementary reasoning paths in in-context versus out-of-context settings, respectively. The results demonstrate that the in-context setting reduces the occurrence of divergence and improves the accuracy of selection. Moreover, the divergence observed between the two methods suggests that LLMs may exhibit an inherent "duality" in their reasoning behaviors.

**Analysis of Divergent Reasoning Path.** To better analyze the reliability of PL-based and NL-based methods when they produce divergent outputs, we propose a new set of evaluation metrics for comparison. The confidence score $C$ is computed using the following formula:

$$C_x = \frac{DC_x + DE_x}{DC + DE}, x \in [NL, PL]. \tag{5}$$

Table 3 shows the main results of confidence score. From a methodological perspective, Self-Dual demonstrates higher reliability than Self-Dual-Auto when divergences occur. For example, in the Deepseek model, Self-Dual achieves 80% confidence compared to Self-Dual-Auto's 69.45%. This indicates that the in-context setting in Self-Dual simultaneously enhances the confidence of both reasoning methods while reducing the likelihood of divergence (from 81% to 68%). Besides, both models consistently show that the NL-based method exhibits higher reliability; however, the contributions of the PL-based method remain significant and should not be overlooked.

**Refinement Strategy Evaluation** In this section, we propose new metrics to evaluate the selection accuracy of each method during the refine stage. Since CE indicates that both solutions are incorrect, we do not consider this case when evaluating the impact on the second stage. The *Selection Correctness Ratio (SCR)* measures the proportion of correct selections made among avoidable errors. It is defined by the following formula:

$$SCR = \frac{CC + DC}{CC + DC + DE}. \tag{6}$$

The *Discordant Correctness Ratio (DCR)* is calculated using the following Equation 7 and is designed to measure the proportion of making the correct selection when the two methods produce divergent answers. It serves as a reliable indicator of the effectiveness of the refine stage.

$$DCR = \frac{DC}{DC + DE}. \tag{7}$$

As shown in Table 4, it can be concluded that Self-Dual performs the best, followed by Self-Dual-Auto, while Auto shows the weakest performance based on the overall data. The difference between Self-Dual-Auto and Auto can be entirely attributed to the design of the refine stage, demonstrating that our proposed refinement strategy is significantly superior to the simple selection mechanism used in Auto. The results on Self-Dual and Self-Dual-Auto also merit further discussion. This may result from the in-context setting, which enhances consistent correctness and supports better decision-making when divergences arise.

Table 5: Main training results on MATH500, AMC23, AIME24 and AIME25. Bold numbers indicate the best accuracy, and underline indicates the second best.

| Model | Backbone Model | RLVR | DataSize | Training Format |
|---|---|---|---|---|
| Qwen2.5-7B-Ins | Qwen2.5-7B-Base | No | - | - |
| *w/ SFT* | Qwen2.5-7B-Ins | No | 3.1k | Self-Dual |
| *w/ SFT+GRPO* | Qwen2.5-7B-Ins | Yes | 7.5k | Self-Dual |
| Qwen2.5-Math-7B-Ins | Qwen2.5-Math-7B-Base | Yes | 3000k | CoT & TIR |

| | MATH500 | AMC23 | AIME24 | AIME25 |
| | *Pass@1* | *Best@64* | *Best@64* | *Best@64* |
|---|---|---|---|---|
| Qwen2.5-7B-Ins | 72.2 | **65.0** | 20.0 | 16.67 |
| *w/ SFT* | 74.4 | 62.5 | 26.67 | 16.67 |
| *w/ SFT+GRPO* | 76.8 | 62.5 | **30.0** | **26.67** |
| Qwen2.5-Math-7B-Ins | **77.8** | 62.5 | 20.0 | 20.0 |

## 4.2 TRAINING-TIME

### 4.2.1 EXPERIMENTAL SETUP

**Training Setup.** We adopt Qwen2.5-7B-Instruct as the backbone model, which is one of the most widely used open-source instruction models. To evaluate the quality of model outputs, we employ Qwen3-32B as the reward model, responsible for computing the format reward. All training data are drawn from the 7.5K training set of the MATH benchmark.

**SFT Data Construction.** We follow the data generation paradigm established in prior work (Guan et al., 2025). Specifically, we use DeepSeek-V3.1 as the generator and score the outputs with Qwen3-32B under the same prompts used in the RL stage. Starting with 3.5K samples as the candidate pool, a sample is included in the SFT set if it satisfies two conditions: the final answer is correct and the format reward exceeds 0.8. Otherwise, the sample remains in the pool and a new answer is generated. We allow up to three iterations and collect approximately 3.1K high-quality SFT samples. During this process, we apply the Self-Dual reasoning template to ensure consistent formatting. In both SFT and RL training, we only provide the problem statement as input to reduce inference latency.

**RLVR Data Construction and Training.** Samples that fail the filtering process are combined with the remaining 4K examples to form a 4.4K RLVR dataset, which encourages exploration in the RL stage. For training, we use the LLaMA-Factory framework for SFT and the VERL framework to perform reinforcement learning with standard GRPO. For SFT, we adopt LoRA with rank 32 and $\alpha = 64$, using a batch size of 32 and a learning rate of $5 \times 10^{-5}$. For GRPO, we train the full model with a batch size of 64 and a learning rate of $1 \times 10^{-6}$. We set the group number $n = 6$ and train for 3 epochs, which corresponds to about 200 steps. All training is conducted on NVIDIA H200 GPUs.

### 4.2.2 MAIN RESULTS

As shown in Table 5, we evaluate the effectiveness of Self-Dual during training by comparing it with Qwen2.5-Math-7B-Instruct, a strong math-specific baseline. Qwen2.5-Math-7B-Instruct is trained on large-scale CoT–TIR mixed data and RLAR, making it a competitive reference point. We focus on the AIME25 benchmark for two reasons. First, AIME25 has high difficulty and can better reflect a model's reasoning ability. Second, both Qwen2.5-Math-7B-Instruct and Qwen2.5-7B-Instruct were released after the creation of AIME25, while our training data come exclusively from the 7.5K MATH training set, ensuring no data contamination. Unlike prior work, Self-Dual does not rely on large hybrid datasets. Instead, it enhances reasoning by explicitly combining complementary paradigms. For this reason, we directly use Qwen2.5-7B-Instruct as the backbone to leverage its existing natural language reasoning and coding capabilities. This setup allows us to isolate the contribution of the Self-Dual framework during training. We further compare Qwen2.5-7B-SD against prior methods on MATH-500 and GSM8K, with results presented in Table 6.

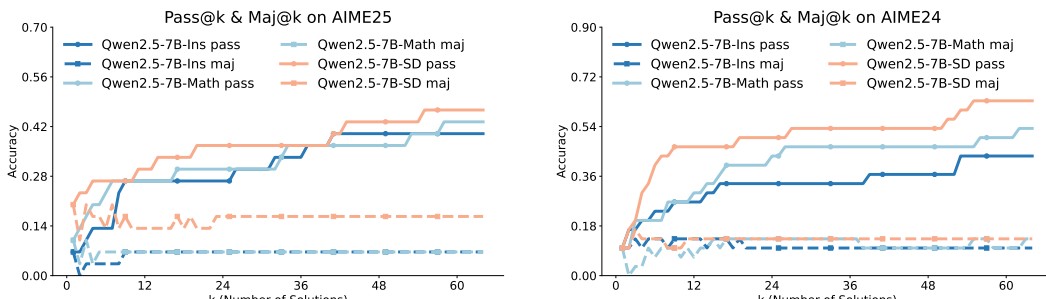

Figure 2: Pass@K and Maj@K curves of Qwen2.5-7B-SD, Qwen2.5-7B-Instruct, and Qwen2.5-7B-Math-Instruct on the AIME24 and AIME25 benchmarks.

We observe that Self-Dual training consistently improves performance across multiple math benchmarks. On MATH500, Qwen2.5-7B-Instruct achieves 72.2, while adding Self-Dual SFT raises the score to 74.4. With GRPO, the score further increases to 76.8, which is close to Qwen2.5-Math-7B-Instruct at 77.8. On AIME24 and AIME25, Self-Dual with SFT and GRPO, denoted as Qwen2.5-7B-SD, achieves 30.0 on AIME24 and 26.67 on AIME25, significantly higher than both the instruct and math baselines. These results show that Self-Dual not only improves single-answer accuracy but also provides stronger generalization on challenging reasoning tasks.

The comparison also highlights the efficiency of Self-Dual. Qwen2.5-Math-7B-Instruct relies on about 3000k mixed samples combining CoT and TIR, whereas Qwen2.5-7B-SD uses only the 7.5k MATH dataset. Despite this gap in scale, Qwen2.5-7B-SD reaches similar accuracy on MATH500 and even surpasses the math-specific model on AIME. This demonstrates that combining complementary reasoning with reinforcement learning can achieve strong results with much less data. We hope that the data construction approach of Self-Dual can inspire further efforts to enhance reasoning ability through data-centric methods.

### 4.3 PASS@K AND MAJ@K

As show in Fig. 2, Self-Dual training yields clear performance gains. On both AIME24 and AIME25, the trained model achieves higher Pass@k than the baselines, including Qwen2.5-7B-Ins and Qwen2.5-7B-Math-Ins. The improvement is especially strong on AIME24, where the model shows higher accuracy at small $k$ and maintains the lead as $k$ increases. This indicates that Self-Dual improves not only single-solution quality but also overall exploration. Another interesting observation comes from the Maj@k curves. On AIME25, the Self-Dual model achieves a stable improvement, reaching about twice the accuracy of the baselines. This indicates that Self-Dual consistently increases the reliability of reasoning by integrating diverse paths. On AIME24, the gains are smaller, but the curve is more stable compared to both instruct and math-instruct baselines. This suggests that Self-Dual improves robustness and consistency, even in settings where performance improvements are less pronounced.

## 5 CONCLUSION

We presented Self-Dual, a framework that unifies NL and PL reasoning by generating complementary trajectories and integrating them through structured self-reflection. At inference time, Self-Dual consistently improves reasoning performance across benchmarks, showing that complementarity can be effectively exploited within a single forward pass. At training time, we further evaluate Self-Dual as a data construction strategy. On the challenging AIME25 benchmark, the resulting model Qwen2.5-7B-SD surpasses both the general instruct and math-specialized baselines, demonstrating the potential of Self-Dual to enhance reasoning under limited data conditions.

## ETHICS STATEMENT

This work focuses on improving the mathematical reasoning capabilities of large language models through the integration of complementary reasoning paradigms. All experiments are conducted on publicly available benchmark datasets such as MATH500, AIME, and GSM8K, which contain no sensitive or personally identifiable information. Our methods do not involve the collection of new human data, nor do they raise direct privacy concerns. The proposed framework is intended to advance fundamental research in reasoning and transparency, and has no foreseeable immediate negative societal impact. Nevertheless, as with all improvements to language models, there remains the potential risk of misuse in high-stakes applications. We encourage responsible deployment, including careful evaluation of robustness and fairness, before applying these methods beyond academic research.

## REPRODUCIBILITY STATEMENT

All datasets used in this work are publicly available. Detailed experimental settings, including model configurations, training procedures, and evaluation protocols, are provided in the paper. We will release our code, data construction scripts, and trained checkpoints to facilitate full reproducibility.

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

## A  COMPLEMENTARY EXPERIMENTS

### A.1  PASS@K ON MATH500

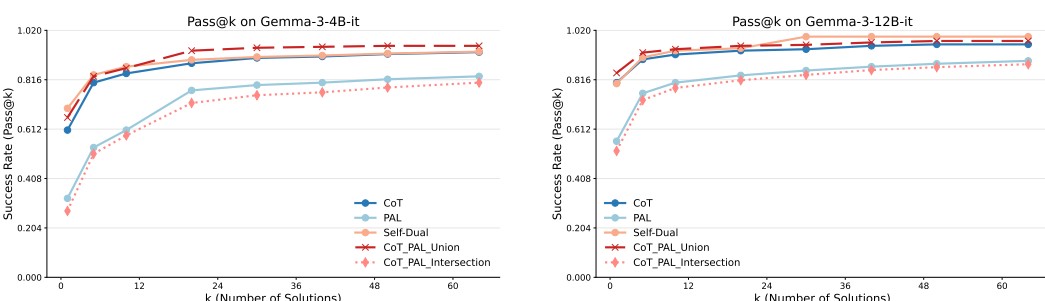

Figure 3: Pass@k curves of Gemma3-4B-it and Gemma3-12B-it on CoT, PAL, Self-Dual, as well as the union and intersection of CoT and PAL, evaluated on the MATH500 dataset.

The results on MATH500 show clear differences among CoT, PAL, Self-Dual, and their combinations. CoT and PAL achieve moderate Pass@k, but both plateau early, indicating limited exploration. Self-Dual consistently outperforms CoT and PAL across different values of $k$, which demonstrates its ability to integrate complementary reasoning paths within a single inference.

The union of CoT and PAL sets provides the highest Pass@k across all $k$, confirming that the two paradigms cover different solution spaces. The intersection curve is lower but stable, reflecting their shared consistency. Importantly, Self-Dual tracks close to the union curve, suggesting that the framework approximates the exploration benefit of multiple samples with only one inference. These results highlight the effectiveness of Self-Dual in leveraging complementarity while maintaining efficiency.

### A.2  ABLATION STUDY

**Contribution of the Initial Stage.** The method DCT in Figure 4 and Table 2 refers to Double CoT in Self-Dual, where two identical CoT-based reasoning paths are used in the initial stage instead of complementary thinking modes. We designed this method to maintain the similar reasoning stages and token consumption as Self-Dual, ensuring a fair comparison. As shown in Figure 4, Self-Dual consistently outperforms both CoT and DCT across the two models. For example, on MATH-500, DCT slightly outperforms CoT by 2.4% on Gemma-3-4B-it, suggesting that iterative reasoning and reflection over dual outputs provide modest performance gains. However, Self-Dual achieves a 5.6% improvement over DCT, highlighting the effectiveness of leveraging complementary reasoning modes across natural and programming languages.

**The Impact of Refine Stage.** Since both Self-Dual-Auto and Auto are built upon existing NL-based and PL-based methods, we ensure that they utilize the same underlying methods and results for consistency. On the diverse MATH-500 dataset, Self-Dual-Auto significantly outperforms Auto in terms of accuracy, as shown in Table 1. For example, on Gemma-3-4B-it, Auto yields a -3.4% change while Self-Dual-Auto achieves a +4.6% improvement. These results suggest that the refinement stage in Self-Dual-Auto is more stable and adaptable than the simple selection mechanism

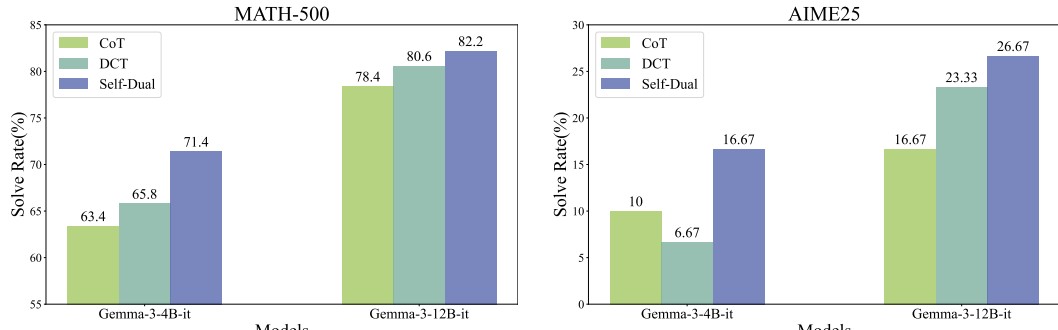

Figure 4: Results of CoT, DCT, and Self-Dual with Gemma-4-4B-it and Gemma-3-12B-it on MATH-500 (a) and AIME (b).

employed by Auto. On the AIME benchmark, except for Gemma-3-4B-it which outperforms Auto, all other models maintain comparable performance. This may suggest that the complementary potential between independently applied CoT and PAL has reached its upper bound on the AIME2025 dataset. The result that BAND outperforms both Self-Dual-Auto and Auto further demonstrates that the Self-Dual format is a more efficient way to combine complementary thinking modes.

### A.3 REMAINING RESULTS

Table 6 presents the remaining results on the GSM8K and SVAMP benchmarks. Consistent with the findings on MATH-500 and AIME, Self-Dual continues to demonstrate superior performance on GSM8K. Notably, even when using Gemma-3-27B-it, Self-Dual without external tools outperforms all other methods, including BAND itself. This advantage may be attributed to inherent randomness and the relatively lower difficulty of the GSM8K dataset. Furthermore, it can be observed that once accuracy surpasses 93%, the performance gains from prompt-based methods over standard approaches become marginal. This plateau may be constrained by the model's inherent capabilities and limitations.

Table 6: The remaining results on SVAMP and GSM8K.

| Methods | GSM8K | | | SVAMP | | | #Call |
|---|---|---|---|---|---|---|---|
| | **G-4B** | **G-12B** | **G-27B** | **G-4B** | **G-12B** | **G-27B** | |
| | *NL-based methods* | | | | | | |
| CoT | 77.33 | 91.96 | 95.15 | 87.44 | 94.11 | **94.67** | 1 |
| Reflextion | 78.24 | 92.12 | 94.09 | 81.33 | 93 | 93.67 | 3 |
| | *PL-based methods* | | | | | | |
| PAL | 70.2 | 91.05 | 93.93 | 83.33 | 91.33 | 92 | 1 |
| TIR | 66.34 | 87.95 | 91.58 | 85.33 | 93.33 | 91.33 | 1-5 |
| *w/o tool* | 13.19 | 26.54 | 41.85 | 65 | 79.33 | 84 | 1 |
| CRITIC | 72.4 | 91.28 | 93.4 | 86 | 92 | 92 | 9 |
| *w/o tool* | 70.96 | 90.9 | 92.95 | 85.67 | 92 | 91.33 | 9 |
| | *Hybrid-based methods* | | | | | | |
| Automatic | 77.86 | 91.48 | 95.25 | 85.67 | 94.33 | 93.67 | 3 |
| SD-Auto | 82.71 | 93.71 | 95.68 | **90** | **94.67** | 93.33 | 3 |
| Self-Dual | **83.07** | 93.61 | 95.43 | 88 | 92 | 92.33 | 2 |
| *w/o tool* | 82.26 | **93.93** | **95.83** | 87.67 | 92 | 91 | 1 |

In contrast to previous results, Self-Dual Auto achieves the best performance on the SVAMP dataset, although its advantage gradually diminishes as model size increases. This trend may be attributed to two main factors: 1) SVAMP is relatively simple and imposes minimal demands on complex computation; and 2) existing models already possess strong capabilities for solving such problems. Together, these factors limit the effectiveness of complementary thinking modes in further improving

performance. Additionally, the reduced divergence in Self-Dual BAND may constrain its ability to exploit complementary reasoning, potentially accounting for its inferior performance compared to LEGO.

## A.4 DATASET

**SVAMP** is a benchmark dataset designed to evaluate LLMs on applied mathematics problems typically encountered in elementary and middle school. The test set of **GSM8K**(Cobbe et al., 2021) contains 1.2K high-quality, linguistically diverse grade school math word problems. **MATH**(Hendrycks et al., 2021) is a large-scale dataset consisting of 5K challenging competition-level mathematics problems. **MATH500** (Lightman et al., 2024) is a curated subset of the MATH dataset, containing 500 high-quality problems, and is commonly used to evaluate the mathematical reasoning capabilities of large language models. **AIME25** comprises competition-level problems from the 2025 American Invitational Mathematics Examination (AIME), including both AIME 2025-I and AIME 2025-II.

Table 7: The details of datasets.

| Dataset | Num | Example | Difficulty | License |
|---|---|---|---|---|
| SVAMP | 300 | He then went to see the oranges being harvested. He found out that they harvest 66 sacks per day and that each sack contains 25 oranges. How many oranges will they have after 87 days of harvest? | Combinations of basic arithmetic operations. | MIT |
| GSM8K | 1319 | A new program had 60 downloads in the first month. The number of downloads in the second month was three times as many as the downloads in the first month, but then reduced by 30% in the third month. How many downloads did the program have total over the three months? | Grade-school level math word problems. | MIT |
| MATH | 5000 | How many fractions in the form $\frac{n}{99}$, with $0 < n < 99$, are in lowest terms? | High school level mathematics competitions. | MIT |
| MATH500 | 500 | The polynomial $x^3 - 3x^2 + 4x - 1$ is a factor of $x^9 + px^6 + qx^3 + r$. Enter the ordered triple $(p, q, r)$. | Subset extracted from the MATH dataset. | MIT |
| AMC23 | 40 | Positive real numbers $x$ and $y$ satisfy $y^3 = x^2$ and $(y - x)^2 = 4y^2$. What is $x + y$? | 2023 American Mathematics Competition, released after February 2023 | - |
| AIME24 | 30 | Find the largest possible real part of $(75 + 117i)z + \frac{96+144i}{z}$ where $z$ is a complex number with $|z| = 4$. | 2024 American Invitational Mathematics Examination, released after February 2024 | - |
| AIME25 | 30 | Quadratic polynomials $P(x)$ and $Q(x)$ have leading coefficients 2 and $-2$, respectively. The graphs of both polynomials pass through the two points $(16, 54)$ and $(20, 53)$. Find $P(0) + Q(0)$. | 2025 American Invitational Mathematics Examination, released after February 2025 | - |

## A.5 BASIC STATISTICS OF SAMPLED RESULTS

Table 8 presents a detailed breakdown of categories within the sampled results. *D2E* refers to cases where the two answers generated during the initial stage are different, and both are incorrect. In addition, the calculation formulas used to derive the results in Table 2 are also provided.

$$DC = DC_{NL} + DC_{PL}. \tag{8}$$

$$DE = D2E + DE_{NL} + DE_{PL}. \tag{9}$$

Table 8: The most detailed categorization of sampled information.

| Models | Methods | DC | CE | D2E | $DC_{NL}$ | $DE_{NL}$ | $DC_{PL}$ | $DE_{PL}$ | Badcode | FN | No2Paths |
|---|---|---|---|---|---|---|---|---|---|---|---|
| Deepseek -V3 | BAND | 77% | 4% | 4% | 5% | 0% | 1% | 2% | 4% | 1% | 4% |
| | LEGO | 67% | 1% | 1% | 7% | 0% | 1% | 2% | 15% | 0% | - |
| | Auto | 67% | 1% | 1% | 3% | 4% | 1% | 2% | 15% | 0% | - |
| Gemma-3 -12B-it | BAND | 67% | 6% | 6% | 8% | 0% | 0% | 2% | 4% | 1% | 9% |
| | LEGO | 53% | 4% | 4% | 14% | 1% | 2% | 4% | 12% | 2% | - |
| | Auto | 53% | 4% | 4% | 10% | 5% | 1% | 5% | 12% | 2% | - |
| | DCT | 80% | 9% | 9% | 3% | 0% | - | - | - | 2% | 4% |

Table 9: Pass@1 accuracy of models in the zero-shot setting on MATH and GSM8k.

| Model | Base | MATH ZS Pass@1 | GSM8k ZS Pass@1 |
|---|---|---|---|
| Skywork-Math (Zeng et al., 2024) | Llama-2 | 47.7 | 72.9 |
| Xwin-Math(Li et al., 2024a) | Llama-2 | 40.6 | 82.6 |
| WizardMath-Qwen (Luo et al., 2025b) | Qwen2.5-Math | 77.8 | 93.9 |
| WizardMath-Qwen (Luo et al., 2025b) | Qwen2.5 | 74.5 | 94.0 |
| WizardMath-DeepSeek (Luo et al., 2025b) | DeepSeekMath | 64.6 | 91.0 |
| DART-Math (Tong et al., 2024b) | DeepSeekMath | 52.9 | 88.2 |
| DeepSeekMath-Instruct-7B (Shao et al., 2024) | DeepSeekMath | 46.8 | 73.6 |
| DeepSeekMath-RL-7B (Shao et al., 2024) | DeepSeekMath | 51.7 | 88.2 |
| MMIQC (Liu et al., 2024) | DeepSeekMath | 45.3 | 79.0 |
| KPMath-Plus (Huang et al., 2024) | DeepSeekMath | 52.9 | 88.2 |
| MetaMath-Llemma-7B (Yu et al., 2024) | Llemma-7B | 30.0 | 69.2 |
| MetaMath-Mistral-7B (Yu et al., 2024) | Mistral-7B | 28.2 | 77.7 |
| MathCoder2-DeepSeekMath (Lu et al., 2024) | DeepSeekMath | 38.6 | 68.8 |
| MathCoder2-Code-Llama (Lu et al., 2024) | Code-Llama | 28.8 | 52.3 |
| MathCoder2-Mistral (Lu et al., 2024) | Mistral | 36.7 | 68.2 |
| ToRA (Gou et al., 2024b) | LLaMA-2 | 40.1 | 68.8 |
| ToRA-CODE (Gou et al., 2024b) | CodeLLaMA | 44.6 | 72.6 |
| NuminaMath-CoT (Li et al., 2024b) | DeepSeekMath | 55.2 | 75.4 |
| CoR-Math-7B (Yu et al., 2025b) | DeepSeekMath | 66.7 | 88.7 |
| Qwen2.5-7B-Math-Instruct (Yang et al., 2024a) | Qwen2.5-7B-Math | **83.6** | **95.2** |
| Qwen-2.5-7B-SD | Qwen2.5-7B-Instruct | 78.12 | 92.4 |

## A.6 INFERENCE-TIME CASE STUDY

We further discuss notable and illustrative examples in the process of reviewing the experimental results in this section.

**Overcoming the Consensus Fallacy.** In general, if both methods reach a consensus during the initial stage, the refine stage typically accepts this agreed-upon answer as the final correct solution. This observation is also evident from the sampled results presented in Table 2. But we also discover an interesting case worth discussing. The two solutions in the first stage consistently arrived at the incorrect answer of 1. However, during the refine stage, the model reanalyzed the problem and successfully produced the correct answer of $\frac{1}{4}$. We believe that the key component that triggers the realization of the initial error lies in the *Look Back* phase—specifically, the prompt segment: *"Let's analyze the problem."* This reflective step encourages the model to revisit and reassess the reasoning process, ultimately enabling it to identify and correct earlier mistakes.

**Reasoning, Verification and Reflection Format.** We also identify a typical case pattern with potential for further utilization: the PL-based method serves to verify the correctness of the result generated by the NL-based method, rather than producing an entirely separate reasoning path. This

format can improve the computational accuracy of the NL-based method, but it comes at the cost of losing the potential benefits of independent PL-mode reasoning.

**Diverse Reasoning, Shared Conclusion.** Another meaningful Case is selected from the CC-type results. Concretely, both solutions correctly identify 27 as the smallest positive perfect cube. The NL-based solution offers a clear and logically structured mathematical explanation, while the PL-based solution validates the answer through straightforward brute-force computation. This case also exemplifies how complementary reasoning modes can manifest in practice.

**Hallucination in Self-Dual.** The sampled results from the Self-Dual method reveal several cases of hallucination, which we report herein. A case illustrates a hallucination in which the LLM fabricates a programming solution and evaluates it as if it were valid, despite the absence of real code. Another case demonstrates that the model mistakenly treats numerically equivalent values expressed in LaTeX format and decimal notation as unequal. A third case involves an incorrect prediction of the code execution result. We hope these cases may inspire future research on hallucination.

### A.7 COMPARE WITH OTHER METHODS

Table 9 summarizes pass@1 results on GSM8k and MATH in the zero-shot setting. All compared models are standardized to 7B parameters to ensure fairness. Our proposed model, Qwen-2.5-7B-SD, achieves 78.1% on MATH and 92.4% on GSM8k. This performance surpasses the expert model on arithmetic reasoning and is second only to the proprietary Qwen-2.5-7B-Math-Instruct on MATH. On GSM8k, our method also outperforms most fine-tuned and RL-based approaches. These results demonstrate that combining NL and PL supervision, even with only 3k samples for SFT initialization and 4k samples for RL training, can substantially enhance reasoning performance under a lightweight training regime.

## B THE USAGE OF LARGE LANGUAGE MODELS

LLMs were used in a limited capacity for brainstorming, minor phrasing suggestions, and partial writing refinement. All generated text was carefully reviewed and substantially revised by the authors. No LLMs were involved in research design, model development or experiments. Therefore, we confirm that LLMs did not play a significant role and should not be regarded as contributors.

## C PROMPTS

Listing 1: The Self-Dual prompt designed for MATH-500 benchmark.

```
You need to follow the following process to solve the math problem
1. Natural language based solution process, i.e., think step by step
    using natural language
2. Programming language based solution process, i.e. writing python code
    to accomplish step-by-step thinking
3. Then you need to scrutinize the sovling step of each solution and
    compare them with each other. Give a new correct solution. You need
    to pay attention to a few points:
   1. please note that you need to ensure that the final answer is the
       correct answer to the user's problem
   2. The answers to the two solutions may not be the same, and you need
       to double-check to get the right answer.
   3. the output of the program is through the python interpreter, so you
        can trust the results of the code execution. So you need to pay
       attention to whether the code solves the problem or not.
   4. Give your answer by '\boxed{}'.

---

Question: Suppose $d\\not=0$. We can write $\\left(12d+13+14d^2\\right)
    +\\left(2d+1\\right)$, in the form $ad+b+cd^2$, where $a$, $b$, and
    $c$ are integers. Find $a+b+c$.
```

Answer:
Natural language based solution process:
12d+13+14d^2 + 2d+1 = 14d^2 + 14d + 14.\nThen $a=14$, $b=14$, $c=0$.\n$a+
    b+c = 14+14+0 = 28$. The answer is 28.

Programming language based solution process:
```python
from sympy import symbols, simplify

d = symbols('d')
expr = (12*d + 13 + 14*d**2) + (2*d + 1)
simplified_expr = simplify(expr)

a = simplified_expr.coeff(d)
b = simplified_expr.coeff('d')
c = simplified_expr.coeff(d**2)

result = a + b + c
print(result)
```
```output
42
```

Scrutinize:
### Analysis of Solutions

#### Natural Language Solution:
1. **Combining Like Terms**:
   - Correctly combines: (12d + 13 + 14d) + (2d + 1)
   - Results in: 14d + (12d + 2d) + (13 + 1) = 14d + 14d + 14
2. **Identifying Coefficients**:
   - Correctly identifies:
    - a (coefficient of d) = 14
    - b (constant term) = 14
    - c (coefficient of d) = 14 (not 0 as stated)
3. **Calculation Error**:
   - Incorrectly states c = 0 (should be 14)
   - Incorrect sum: 14 + 14 + 0 = 28 (should be 14 + 14 + 14 = 42)

#### Programming Solution:
1. **Symbolic Computation**:
   - Correctly uses sympy to simplify the expression
   - Gets correct simplified form: 14d + 14d + 14
2. **Coefficient Extraction**:
   - Correctly extracts coefficients:
    - a (d term) = 14
    - b (constant) = 14
    - c (d term) = 14
3. **Correct Sum**:
   - Computes 14 + 14 + 14 = 42

### Key Differences:
1. The natural language solution made an error in identifying c (
    coefficient of d) as 0 when it should be 14
2. The programming solution correctly handles all coefficients through
    symbolic computation

### Corrected Solution:
1. Combine like terms:
   (12d + 13 + 14d) + (2d + 1) = 14d + 14d + 14
2. Identify coefficients:
   - a (d) = 14
   - b (constant) = 14
   - c (d) = 14

```
3. Compute sum:
   a + b + c = 14 + 14 + 14 = 42
```

### Final Answer:
The correct answer is $\boxed{42}$.

The programming solution was correct, while the natural language solution
    contained an error in identifying the d coefficient. The proper sum
    of coefficients is indeed 42.

---

Question: How many combinations of pennies (1 cent), nickels (5 cents)
    and/or dimes (10 cents) are there with a total value of 25 cents?

Answer:
Natural language based solution process:
1. Let $p$ be the number of pennies, $n$ the number of nickels and $d$
    the number of dimes. We want to find the number of non-negative
    integer solutions to $p + 5n + 10d = 25$.

We can consider cases based on the number of dimes $d$.
\\begin{itemize}
   \\item $d=0$: $p+5n=25$. Then $5n \\le 25$, so $n \\le 5$. $n$ can take
        values from 0 to 5, so there are 6 solutions.
   \\item $d=1$: $p+5n=15$. Then $5n \\le 15$, so $n \\le 3$. $n$ can take
        values from 0 to 3, so there are 4 solutions.
   \\item $d=2$: $p+5n=5$. Then $5n \\le 5$, so $n \\le 1$. $n$ can take
        values from 0 to 1, so there are 2 solutions.
\\end{itemize}\
The total number of solutions is $6+4+2=12$.
Alternatively, we can express $p = 25 - 5n - 10d$. Since $p \\ge 0$, we
    have $25 - 5n - 10d \\ge 0$, so $5n + 10d \\le 25$, which simplifies
    to $n + 2d \\le 5$.
For $d=0$, $n \\le 5$, so $n=0,1,2,3,4,5$ (6 solutions).
For $d=1$, $n \\le 3$, so $n=0,1,2,3$ (4 solutions).
For $d=2$, $n \\le 1$, so $n=0,1$ (2 solutions).
For $d=3$, $n \\le -1$, which is impossible.
Thus the total number of solutions is $6+4+2=12$.

The answer is 12.

Programming language based solution process:
```python
def solve():
  count = 0
  for p in range(26):
    for n in range(5):
      for d in range(5):
        if 1*p + 5*n + 10*d == 25:
          count += 1
  print(count)
solve()
```
```output
11
```

Scrutinize:
### Analysis of Solutions

#### Problem Interpretation:
We need to count all non-negative integer solutions to the equation:
p + 5n + 10d = 25
where:

```
- p = number of pennies (1)
- n = number of nickels (5)
- d = number of dimes (10)

#### Evaluating the Solutions:

1. **Natural Language Solution**:
   - Correctly analyzes cases by dimes (d):
     - d=0: 6 solutions (n=0 to 5)
     - d=1: 4 solutions (n=0 to 3)
     - d=2: 2 solutions (n=0 to 1)
     - d>=3: No solutions
   - Total: 6 + 4 + 2 = 12
   - *Correct*: Systematic and complete enumeration

2. **Programming Solution**:
   - Uses brute-force enumeration:
     - p: 0-25
     - n: 0-4
     - d: 0-2 (should be 0-2, as written)
   - Outputs 11 (incorrect)
   - *Issue*: The range for n should be 0-5 (not 0-4) since 5 nickels make
       25

#### Corrected Programming Solution:
```python
def count_combinations():
    count = 0
    for d in range(0, 3): # dimes: 0-2
        for n in range(0, 6): # nickels: 0-5
            p = 25 - 5*n - 10*d
            if p >= 0:
                count += 1
    print(count)

count_combinations()
```
```output
12
```

### Final Answer:
There are $\boxed{12}$ valid combinations of coins that sum to 25 cents.
```

Listing 2: The reward prompt designed for filtering and GRPO training.

```
You are tasked with evaluating a response to a math problem to determine
    if it adheres to the specified format and requirements. You must
    assign a score between 0 and 1 based on the response's compliance
    with the format, focusing on completeness and clarity/structure. Your
     evaluation must check each component, identify deviations, provide
    suggestions for improvement, and output the results in a structured
    format that includes the natural language (NL) and programming
    language (PL) answers, as well as the assigned score. If the natural
    language solution is missing, report `[NL-Based Answer]` as `None`.

### Expected Format and Requirements
The response must solve a math problem using the following structure:
1. Natural Language-Based Solution Process:
   - Provide a step-by-step explanation in natural language, solving the
       problem as a human would.
   - Clearly combine terms, derive coefficients, or perform calculations
       as needed.
   - Identify the final values (e.g., coefficients a, b, c) and compute
       the required answer.
```

2. Programming Language-Based Solution Process:
   - Provide Python code that solves the problem step-by-step using
     symbolic computation (e.g., with `sympy`) or numerical methods.
   - The code must include a print statement to output the final result.
   - The code must be executable through a Python interpreter.
3. Scrutiny and Comparison:
   - Analyze both the natural language and programming solutions.
   - Compare the two solutions to determine if they align or differ in
     their approach or results.
   - If discrepancies exist, explain why and provide a reconciled solution
     .
4. Final Answer:
   - Present the final answer in a LaTeX box format: `\\boxed{{}}`.
   - The response must terminate after the `\\boxed{{}}` answer, with no
     additional or redundant content.
5. Additional Notes:
   - The response must explicitly label sections (e.g., "Natural Language-
     Based Solution Process," "Programming Language-Based Solution
     Process," "Scrutiny," "Final Answer").
   - The programming solution must be verifiable via Python execution.
   - The scrutiny section must compare the two approaches and explain any
     differences.
   - The response must be coherent, with logical and organized reasoning
     in both the natural language and scrutiny sections.

### Scoring Criteria
Assign a score between 0 and 1 based on the following criteria:

- Completeness (60%):
  - 15% for including a natural language solution.
  - 15% for including a complete, executable Python code solution.
  - 15% for including a scrutiny section that compares both solutions.
  - 15% for including a final answer in `\\boxed{{}}` format.
- Clarity and Structure (40%):
  - 20% for clear, step-by-step, and coherent explanations in the natural
    language solution (if present), with logical progression and no
    rambling or disorganized reasoning.
  - 10% for proper labeling of sections (e.g., "Natural Language-Based
    Solution Process").
  - 10% for a thorough scrutiny section that clearly compares solutions
    and explains differences in an organized manner.
- Penalties:
  - Deduct up to 15% per missing section (e.g., no natural language
    solution, no scrutiny section).
  - Deduct up to 10% for minor issues (e.g., unclear steps, improper
    labeling).
  - Deduct up to 20% for major issues (e.g., incomplete code, vague
    scrutiny section).
  - Format Penalty: Deduct up to 20% if the response continues with
    redundant or repeated content after the `\\boxed{{}}` answer (e.g.,
    restarting the solution, adding unnecessary explanations, or
    reiterating the answer).
  - Clarity Penalty: Deduct up to 25% for incoherent, disorganized, or
    rambling reasoning in the natural language or scrutiny sections (e.g
    ., illogical jumps, contradictory statements, excessive verbosity,
    or lack of clear progression).
  - A score of 0 is assigned if the response is entirely absent or
    irrelevant.

### Evaluation Task
Given a response to a math problem, evaluate its compliance with the
   format and assign a score. Provide your assessment in the following
   structure:

1. Compliance Check:

```
    - Does the response include all required sections (Natural Language-
      Based Solution, Programming Language-Based Solution, Scrutiny,
      Final Answer)?
    - Does the natural language solution (if present) follow a clear, step-
      by-step, and coherent process?
    - Does the programming solution include executable Python code?
    - Does the scrutiny section compare solutions and explain differences
      in an organized manner?
    - Is the final answer presented in `\\boxed{{}}`, with no redundant
      content afterward?
2. Issues Identified:
    - List any deviations from the required format (e.g., missing sections,
        incorrect labeling, unclear explanations, non-executable code).
    - Note if the natural language solution is missing or incomplete.
    - Note if the scrutiny section fails to compare solutions or explain
      differences.
    - Note if the response continues with redundant or repeated content
      after `\\boxed{{}}`.
    - Note if the natural language or scrutiny sections are incoherent,
      disorganized, or rambling.
3. Suggestions for Correction:
    - For each issue, suggest how the response could be revised to meet the
        requirements (e.g., add missing section, clarify steps, ensure
      code is executable, terminate after `\boxed{{}}`, improve coherence
      ).
4. Score Breakdown:
    - Provide a breakdown of the score based on Completeness (60%) and
      Clarity and Structure (40%).
    - Justify deductions for any missing components, issues, or penalties (
      including format and clarity penalties).
5. Overall Assessment:
    - State the final score (0 to 1, rounded to two decimal places).
    - Summarize whether the response complies with the format.

### Output Format
Provide your evaluation in the following format:

Evaluation of Response

Compliance Check:
- [ ] All required sections included
- [ ] Natural language solution is step-by-step, clear, and coherent (if
  present)
- [ ] Programming solution includes executable Python code
- [ ] Scrutiny section compares solutions and explains differences in an
  organized manner
- [ ] Final answer in `\\boxed{{}}` with no redundant content afterward

Issues Identified:
- [List specific issues, e.g., "Missing natural language solution," "
  Improper section labeling," "Scrutiny section lacks comparison," "
  Redundant content after `\\boxed{{}}`," "Incoherent reasoning in
  natural language solution."]

Suggestions for Correction:
- [For each issue, provide a specific suggestion, e.g., "Include a
  natural language solution with step-by-step reasoning," "Label
  sections clearly as specified," "Terminate response after `\\boxed
  {{}}`," "Organize reasoning to avoid rambling."]

Score Breakdown:
- Completeness (60%): [Score, e.g., 45/60, with justification, e.g., "
  Missing natural language solution (-15%)"]
```

- Clarity and Structure (40%): [Score, e.g., 20/40, with justification, e
  .g., "Clear code but improper section labeling (-10%), redundant
  content after `\\boxed{{}}` (-20%), incoherent reasoning (-20%)"]
- Total Score: [Score, e.g., 0.65]

[NL-Based Answer]: [Extract the final answer from the natural language
  solution, e.g., "28," or "None" if missing]
[PL-Based Answer]: [Extract the final answer from the programming
  solution, e.g., "42"]
[Format Reward]: [Final score, e.g., 0.65]

Overall Assessment:
- [Summarize compliance, e.g., "The response partially complies due to a
  missing natural language solution, improper section labeling,
  redundant content after `\\boxed{{}}`, and disorganized reasoning.
  The programming solution and scrutiny section are present but need
  clearer comparison and more coherent reasoning."]

### Example Response to Evaluate
[Insert the response to be evaluated here, or provide a placeholder if
  none is given.]

### Notes for the Model
- If no response is provided, indicate that a response must be submitted
  for evaluation and assign a score of 0.
- If the natural language solution is missing, report `[NL-Based Answer]`
  as `None`.
- Ensure the score is calculated systematically based on the criteria and
  rounded to two decimal places.
- Extract the `[NL-Based Answer]` and `[PL-Based Answer]` directly from
  the respective sections of the response, if available.
- If the problem involves symbolic computation (e.g., finding
  coefficients), ensure the programming solution uses appropriate tools
  like `sympy`.
- Do not evaluate the correctness of the answers (e.g., whether the
  coefficients or final answer are mathematically correct); focus
  solely on format compliance, completeness, and clarity/structure.
- Do not modify the response or provide a new solution unless explicitly
  asked; focus on evaluating compliance and scoring.
- Apply the format penalty (up to 20%) for responses that include
  redundant or repeated content after the `\\boxed{{}}` answer, such as
  restarting the solution, reiterating the answer, or adding
  unnecessary explanations.
- Apply the clarity penalty (up to 25%) for responses with incoherent,
  disorganized, or rambling reasoning in the natural language or
  scrutiny sections, such as illogical jumps, contradictory statements,
  excessive verbosity, or lack of clear progression.

### Example Application
Evaluation of Response

Compliance Check:
- [x] All required sections included
- [ ] Natural language solution is step-by-step, clear, and coherent
- [x] Programming solution includes executable Python code
- [x] Scrutiny section compares solutions and explains differences in an
  organized manner
- [ ] Final answer in `\\boxed{{}}` with no redundant content afterward

Issues Identified:
- The natural language solution is present but disorganized, with
  rambling explanations and illogical jumps (e.g., contradictory steps
  in deriving coefficients).
- Section labeling is correct, but the scrutiny section could be more
  concise in explaining differences.

```
     - The response includes redundant content after `\\boxed{{}}`, restarting
         the explanation with an alternative approach.

     Suggestions for Correction:
     - Revise the natural language solution to follow a clear, logical, and
         concise step-by-step process, eliminating rambling or contradictory
         statements.
     - Streamline the scrutiny section to focus on the key discrepancy without
         redundant explanation.
     - Remove the redundant content after `\\boxed{{}}` to adhere to the
         format requirement of terminating the response.

     Score Breakdown:
     - Completeness (60%): 60/60
       - Natural language solution included (15%).
       - Programming solution included and executable (15%).
       - Scrutiny section included (15%).
       - Final answer in `\\boxed{{}}` (15%).
     - Clarity and Structure (40%): 15/40
       - Natural language solution is disorganized and rambling (-15%, clarity
           penalty).
       - Sections are properly labeled (10%).
       - Scrutiny section is thorough but slightly verbose (10%).
       - Redundant content after `\\boxed{{}}` (-15%, format penalty).
     - Total Score: (60 + 15) / 100 = 0.75

     [NL-Based Answer]: 28
     [PL-Based Answer]: 42
     [Format Reward]: 0.75

     Overall Assessment:
     - The response complies with most format requirements, including all
         required sections. However, the natural language solution is
         disorganized and rambling, reducing clarity. The programming solution
          is executable, and the scrutiny section compares approaches but is
         slightly verbose. A significant deduction is made for redundant
         content after `\boxed{{}}` and incoherent reasoning in the natural
         language solution.
     """

     FILTER_PROMPT = """
     You are tasked with evaluating a response to a math problem to determine
         if it adheres to the specified format and requirements. You must
         assign a score between 0 and 1 based on the response's compliance
         with the format, focusing on completeness and clarity/structure. Your
          evaluation must check each component, identify deviations, provide
         suggestions for improvement, and output the results in a structured
         format that includes the natural language (NL) and programming
         language (PL) answers, as well as the assigned score. If the natural
         language solution is missing, report `[NL-Based Answer]` as `None`.

     ### Expected Format and Requirements
     The response must solve a math problem using the following structure:
     1. Natural Language-Based Solution Process:
       - Provide a step-by-step explanation in natural language, solving the
           problem as a human would.
       - Clearly combine terms, derive coefficients, or perform calculations
           as needed.
       - Identify the final values (e.g., coefficients a, b, c) and compute
           the required answer.
     2. Programming Language-Based Solution Process:
       - Provide Python code that solves the problem step-by-step using
           symbolic computation (e.g., with `sympy`) or numerical methods.
```

    – The code must include a prisnt statement to output the final result.
    – The code must be executable through a Python interpreter.
3. Scrutiny and Comparison:
  – Analyze both the natural language and programming solutions.
  – Compare the two solutions to determine if they align or differ in
     their approach or results.
  – If discrepancies exist, explain why and provide a reconciled solution
     .
4. Final Answer:
  – Present the final answer in a LaTeX box format: `\boxed{{}}`.
5. Additional Notes:
  – The response must explicitly label sections (e.g., "Natural Language-
     Based Solution Process," "Programming Language-Based Solution
     Process," "Scrutinize," "Final Answer").
  – The programming solution must be verifiable via Python execution.
  – The scrutiny section must compare the two approaches and explain any
     differences.

### Scoring Criteria
Assign a score between 0 and 1 based on the following criteria:
– Completeness (60%):
  – 15% for including a natural language solution.
  – 15% for including a complete, executable Python code solution.
  – 15% for including a scrutiny section that compares both solutions.
  – 15% for including a final answer in `\boxed{{}}` format.
– Clarity and Structure (40%):
  – 20% for clear, step-by-step explanations in the natural language
     solution (if present).
  – 10% for proper labeling of sections (e.g., "Natural Language-Based
     Solution Process").
  – 10% for a thorough scrutiny section that clearly compares solutions
     and explains differences.
– Penalties:
  – Deduct up to 15% per missing section (e.g., no natural language
     solution, no scrutiny section).
  – Deduct up to 10% for minor issues (e.g., unclear steps, improper
     labeling).
  – Deduct up to 20% for major issues (e.g., incomplete code, vague
     scrutiny section).
  – A score of 0 is assigned if the response is entirely absent or
     irrelevant.

### Evaluation Task
Given a response to a math problem, evaluate its compliance with the
   format and assign a score. Provide your assessment in the following
   structure:

1. Compliance Check:
  – Does the response include all required sections (Natural Language-
     Based Solution, Programming Language-Based Solution, Scrutiny,
     Final Answer)?
  – Does the natural language solution (if present) follow a clear, step-
     by-step process?
  – Does the programming solution include executable Python code?
  – Does the scrutiny section compare solutions and explain differences?
  – Is the final answer presented in `\boxed{{}}`?
2. Issues Identified:
  – List any deviations from the required format (e.g., missing sections,
      incorrect labeling, unclear explanations, non-executable code).
  – Note if the natural language solution is missing or incomplete.
  – Note if the scrutiny section fails to compare solutions or explain
     differences.
3. Suggestions for Correction:

```
        - For each issue, suggest how the response could be revised to meet the
          requirements (e.g., add missing section, clarify steps, ensure
          code is executable).
4. Score Breakdown:
    - Provide a breakdown of the score based on Completeness (60%) and
      Clarity and Structure (40%).
    - Justify deductions for any missing components or issues.
5. Overall Assessment:
    - State the final score (0 to 1, rounded to two decimal places).
    - Summarize whether the response complies with the format.

### Output Format
Provide your evaluation in the following format:

Evaluation of Response

Compliance Check:
- [ ] All required sections included
- [ ] Natural language solution is step-by-step and clear (if present)
- [ ] Programming solution includes executable Python code
- [ ] Scrutiny section compares solutions and explains differences
- [ ] Final answer in `\boxed{{}}`

Issues Identified:
- [List specific issues, e.g., "Missing natural language solution," "
    Improper section labeling," "Scrutiny section lacks comparison."]

Suggestions for Correction:
- [For each issue, provide a specific suggestion, e.g., "Include a
    natural language solution with step-by-step reasoning," "Label
    sections clearly as specified."]

Score Breakdown:
- Completeness (60%): [Score, e.g., 45/60, with justification, e.g., "
    Missing natural language solution (-15%)"]
- Clarity and Structure (40%): [Score, e.g., 30/40, with justification, e
    .g., "Clear code but improper section labeling (-10%)"]
- Total Score: [Score, e.g., 0.75]

[NL-Based Answer]: [Extract the final answer from the natural language
    solution, e.g., "28," or "None" if missing]
[PL-Based Answer]: [Extract the final answer from the programming
    solution, e.g., "42"]
[Format Reward]: [Final score, e.g., 0.75]

Overall Assessment:
- [Summarize compliance, e.g., "The response partially complies due to a
    missing natural language solution and improper section labeling. The
    programming solution and scrutiny section are present but need
    clearer comparison."]

### Example Response to Evaluate
[Insert the response to be evaluated here, or provide a placeholder if
    none is given.]

### Notes for the Model
- If no response is provided, indicate that a response must be submitted
    for evaluation and assign a score of 0.
- If the natural language solution is missing, report `[NL-Based Answer]`
    as `None`.
- Ensure the score is calculated systematically based on the criteria and
    rounded to two decimal places.
- Extract the `[NL-Based Answer]` and `[PL-Based Answer]` directly from
    the respective sections of the response, if available.
```

- If the problem involves symbolic computation (e.g., finding
    coefficients), ensure the programming solution uses appropriate tools
     like `sympy`.
- Do not evaluate the correctness of the answers (e.g., whether the
    coefficients or final answer are mathematically correct); focus
    solely on format compliance, completeness, and clarity/structure.
- Do not modify the response or provide a new solution unless explicitly
    asked; focus on evaluating compliance and scoring.

---

### Example Application
Compliance Check:
- [x] All required sections included
- [x] Natural language solution is step-by-step and clear
- [x] Programming solution includes executable Python code
- [x] Scrutiny section compares solutions and explains differences
- [x] Final answer in `\boxed{{}}`

Issues Identified:
- The natural language solution is present but contains a minor error in
    clarity (incorrectly states c = 0, though steps are clear).
- Section labeling is correct, but the scrutiny section could be more
    concise in explaining differences.

Suggestions for Correction:
- Clarify the coefficient c = 14 in the natural language solution to
    avoid confusion.
- Streamline the scrutiny section to focus on the key discrepancy (c = 0
    vs. c = 14) without redundant explanation.

Score Breakdown:
- Completeness (60%): 60/60
  - Natural language solution included (15%).
  - Programming solution included and executable (15%).
  - Scrutiny section included (15%).
  - Final answer in `\boxed{{}}` (15%).
- Clarity and Structure (40%): 35/40
  - Natural language solution is clear but has a minor error in
     coefficient c (-5%).
  - Sections are properly labeled (10%).
  - Scrutiny section is thorough but slightly verbose (10%).
- Total Score: (60 + 35) / 100 = 0.95

[NL-Based Answer]: 28
[PL-Based Answer]: 42
[Format Reward]: 0.95

Overall Assessment:
- The response fully complies with the format, including all required
    sections. The natural language solution is clear but has a minor
    error in stating c = 0. The programming solution is executable and
    clear, and the scrutiny section effectively compares the two
    approaches. A slight deduction is made for the clarity issue in the
    natural language solution.

---

### Notes on Modifications
- Scoring Structure: Changed to Completeness (60%) and Clarity and
    Structure (40%), removing Correctness as requested. Each section
    under Completeness is weighted equally (15%), and Clarity and
    Structure is split to emphasize the natural language explanation
    (20%) while maintaining weight for labeling and scrutiny (10% each).

```
- NL Missing Case: Explicitly instructs to report `[NL-Based Answer]` as
    `None` if the natural language solution is absent.
- Correctness Removed: The prompt no longer evaluates the mathematical
    accuracy of the answers, focusing solely on format compliance,
    completeness, and clarity.
- Penalties: Adjusted to reflect the new scoring weights, with deductions
    scaled to the 60/40 split.

---

Question: {question}
Model's Response: {response}
Ground Truth: {ground_truth}

Give your detailed Evaluation.
```

