# OpenReview forum: "Self-Dual: Unifying Natural Language and Programmatic Thinking for Enhanced Mathematical Reasoning in LLMs"
_ICLR.cc/2026/Conference — ICLR 2026 Conference Withdrawn Submission_

### Official Review · Reviewer_Pzme · 2025-10-25

**Soundness:** 2
**Presentation:** 3
**Contribution:** 2
**Rating:** 4
**Confidence:** 3

**Summary:**

This paper introduces Self-Dual, a unified framework that integrates natural-language and programmatic reasoning for mathematical problem solving. During inference, the model first generates a natural-language reasoning trajectory and then conditions on it to produce a programmatic trajectory. Finally, a structured refinement stage compares the two paths and synthesizes the final answer. The authors demonstrate the complementarity between the two reasoning modes through Pass@k analyses of their union, intersection, and symmetric difference. Experiments on Gemma-3 and DeepSeek-V3-0324 show consistent improvements on AIME25 and MATH500, while requiring only a small number of LLM calls. Finally, the authors extend the framework to training by constructing Self-Dual-formatted data and applying GRPO to fine-tune Qwen2.5-7B-SD on 7.5K MATH samples.

**Strengths:**

S1: The paper introduces a clear and practical principle of complementarity between NL and PL reasoning.

S2: The Self‑Dual inference design is simple and effective: generate two complementary trajectories in one pass and reconcile them via a three-step refinement that can incorporate interpreter feedback.

S3: The empirical results show consistent gains over CoT, PAL, and several hybrid baselines across model sizes on different benchmarks.

**Weaknesses:**

W1: The introduction in Section 3.1 is unnecessarily long, as the paper reintroduces widely known notions such as Pass@k and basic set operations (union/intersection) that the community already understands.

W2: The method description in Sections 3.2 and 3.3 lacks algorithmic specificity, as the paper provides neither a framework diagram nor step-by-step pseudocode or clear I/O interfaces for the "Dual-Path Generation” and "Refinement” stages, which makes it difficult to reproduce the approach.

W3: The paper introduces a two-stage inference pipeline (NL → PL with a refinement pass), but it does not quantify the token usage, runtime latency, or interpreter overhead; a cost analysis beyond "# of calls” is necessary to assess practicality under real deployment budgets.

W4: The overall idea appears conceptually simple and close to existing multi-stage refinement frameworks (e.g., Self-Refine, CRITIC, TIR, Auto-CoT/selector-based hybrids), which raises questions about novelty.

W5: The experimental section omits several closely related hybrid reasoning baselines (e.g., Program-of‑Thoughts (PoT), MathPrompter, Least‑to‑Most/Tree‑of‑Thought as competitive inference strategies).

W6: The training objective in Section 3.3 leaves several ambiguities, including how the "format reward” is computed, when the binary gate is triggered, and whether intermediate trajectory quality influences optimization.



Minor points:

+ It's suggested to add pseudocode for Dual‑Path Generation and the three‑step refinement, including how execution feedback is injected and how conflicts are resolved to produce the final answer. The presentation would also benefit from a concise methodology figure that shows inputs, prompts, and artifacts flowing through NL and PL paths.
+ The abstract’s baselines on AIME25 (CoT 39.2%, PAL 35.6%) do not match Table 1 (CoT 27.78%, PAL 16.67%) under DeepSeek‑V3‑0324
+ Typos & minor issues. "Reflextion”→"Reflexion” (Table 1); "sovling”→"solving” (Listing 1); "prisnt”→"print” (FILTER_PROMPT); "containes”→"contains” (Table 7 example text); inconsistent use of "BAND/LEGO” vs "Self‑Dual/SD‑Auto” (Appendix A.5).

**Questions:**

See above.

---

### Official Review · Reviewer_unX2 · 2025-10-30

**Soundness:** 2
**Presentation:** 2
**Contribution:** 2
**Rating:** 4
**Confidence:** 4

**Summary:**

The paper introduces the Self-Dual framework to enhance mathematical reasoning in large language models (LLMs) by integrating natural language (NL) and programmatic language (PL) reasoning. It features dual-path generation, which outputs NL and PL reasoning paths simultaneously, and structured self-reflection to reconcile conflicts through a "review-decompose-solve" process. Using complementary reasoning data, the framework achieves efficient few-shot training.
Experiments show the Self-Dual framework excels in benchmarks, achieving 47.8% accuracy on AIME25 and 78.1% on MATH-500, outperforming traditional reasoning paradigms like CoT and PAL. This demonstrates the complementary advantages of NL-PL collaboration.

**Strengths:**

1. The paper introduces a novel approach to combining natural language with programmatic language, showcasing a new method to enhance mathematical reasoning. It achieves this through complementary reflection during the inference phase and by constructing datasets using a self-dual approach during the training phase.
2. The self-dual architecture demonstrates outstanding performance across multiple benchmarks, and Experiments were conducted on models of varying sizes, showing effectiveness in both training and testing phases.

**Weaknesses:**

1. The results of the CoT and PAL union, as shown in Figure 1, seem quite promising, especially with gemma-3-12B-instruct, where self-dual consistently does not surpass it. Given this, what are the advantages of your method compared to directly performing a union? It appears that the union approach may be better in terms of both effectiveness and efficiency, whereas self-dual might be affected by intersection.
2. In Table 5, how does the performance of Qwen2.5-7B-Ins w/ SFT and Qwen2.5-7B-Ins w/ SFT + GRPO compare when not using the method proposed in your paper? How can you demonstrate that the performance improvements are due to your method, rather than the possibility that the higher metrics of w/ SFT + GRPO compared to w/ SFT are primarily due to the inclusion of GRPO?
3. Regarding your focus on the AIME25 benchmark, first, why is AIME25 considered more challenging? Second, since Qwen2.5-math-7B-instruct and Qwen2.5-7B-instruct were released after AIME25, how do you ensure there is no data contamination? Perhaps the argument that there is no data contamination would be more valid if they were released before the construction of the AIME25 dataset.

**Questions:**

Please refer to Weaknesses.

---

### Official Review · Reviewer_8bK2 · 2025-10-30

**Soundness:** 2
**Presentation:** 2
**Contribution:** 2
**Rating:** 2
**Confidence:** 3

**Summary:**

This paper presents Self-Dual, a framework that unifies natural language (NL) and programmatic language (PL) reasoning within large language models (LLMs) to improve mathematical reasoning. The authors demonstrate that NL and PL reasoning are complementary and propose generating dual reasoning trajectories in a single inference pass, followed by a structured self-reflection to integrate these outputs. They also extend the framework to training by creating complementary reasoning datasets. Experiments show that Self-Dual consistently outperforms NL-only, PL-only, and existing hybrid baselines on various math benchmarks, including AIME25 and MATH500. The training experiments suggest that Self-Dual improves LLM performance with significantly less data compared to previous approaches.

**Strengths:**

* This paper proposes a novel unified reasoning framework that explicitly leverages complementarity between NL and PL reasoning in a single inference pass.
* The proposed method demonstrates consistent and significant improvements over strong baselines across multiple benchmarks at both inference and training stages.

**Weaknesses:**

* Experiments are primarily conducted on DeepSeek and Gemma model families; broader validation across other widely-used models (e.g., LLaMA, Qwen) and additional reasoning datasets is lacking.
* The dual-path generation and refinement process requires multiple LLM calls, which increases computational overhead and may hinder real-time deployment.
* The framework and reasoning process are not accompanied by a conceptual diagram or flowchart, which could help readers better understand the dual-path structure and refinement stages.
* The core idea of combining natural language and programmatic reasoning has been explored in prior work such as MathCoder [1], which may limit the perceived novelty of the proposed approach.


Reference

[1] Wang et al. "MathCoder: Seamless Code Integration in LLMs for Enhanced Mathematical Reasoning." ICLR 2024.

**Questions:**

Please refer to "Weaknesses".

---

### Note · Authors · 2025-11-23

I have read and agree with the venue's withdrawal policy on behalf of myself and my co-authors.